

# Impacts of the North Atlantic Oscillation on Winter Precipitations and Storm Track Variability in Southeast Canada and Northeast US

Julien Chartrand[1] and Francesco Salvatore Rocco Pausata[1]

[1]Centre ESCER, Department of Earth and Atmospheric Sciences, University of Quebec in Montreal, Montreal, QC, Canada

**Correspondence:** Julien Chartrand (chartrand.julien.2@courrier.uqam.ca)

**Abstract.** The North Atlantic Oscillation (NAO) affects atmospheric variability from eastern North America to Europe. Although the link between the NAO and winter precipitations in the eastern North America have been the focus of previous work, only few studies have hitherto provided clear physical explanations on these relationships. In this study we revisit and extend the analysis of the effect of the NAO on winter precipitations over a large domain covering southeast Canada and the

northeastern United States. Furthermore, here we use the recent ERA5 reanalysis dataset (1979-2018), which currently has the highest available horizontal resolution for a global reanalysis (0.25°), to track extratropical cyclones to delve into the physical processes behind the relationship between NAO and precipitation, snowfall, snowfall-to-precipitation ratio (S/P), and snow cover depth anomalies in the region. In particular, our results show that positive NAO phases are associated with less snowfall over a wide region covering Nova Scotia, New England and the Mid-Atlantic of the United States relative to

negative NAO phases. Henceforth, a significant negative correlation is also seen between S/P and the NAO over this region. This is due to a decrease (increase) in cyclogenesis of coastal storms near the United States east coast during positive (negative) NAO phases, as well as a northward (southward) displacement of the mean storm track over North America.

## 1 Introduction

The North Atlantic Oscillation (NAO) is the dominant mode of atmospheric variability in the North Atlantic

(Walker, 1925; Walker and Bliss, 1932; Van Loon and Rogers, 1978; Hurrell, 1995; Hurrell et al., 2003). The NAO refers to swings in the atmospheric pressure difference between the Icelandic low and Azores high and is a key factor in the cool-season climate variability from the eastern coast of the United States to Siberia and from the Arctic to the subtropical Atlantic. A common measure of the NAO phase is the so-called NAO index (NAOI) that is determined by the strength and the location of the above-mentioned centers of actions (Walker, 1925; Walker and Bliss, 1932). The NAOI is commonly

defined as the difference in the normalized sea level pressure (SLP) anomalies between Stykkisholmur/Reykjavik, Iceland and either Lisbon, Portugal, or Ponte Delgada, Azores (Hurrell, 1995). Positive NAO phases are associated with a deepening of the Icelandic low and a strengthening of the Azores high. The increased SLP gradient consequently lead to enhanced westerly flow and a northward shift of the mid-latitude storm track (Rogers, 1990; Hurrell and Van Loon, 1997). The



negative phase of the NAO is associated with weaker westerlies, an increase of meandering of the jet stream and higher than
average pressure over Greenland and Iceland.

During positive NAO phases, warmer winter conditions prevail in northern and central Europe and in the eastern United States, and cooler conditions prevail over northern Africa and northeastern Canada. Positive phases are also characterized by above-normal precipitation in northern Europe and drier conditions in the Mediterranean area. The negative phase of the NAO is often characterized by a displacement of the Icelandic low near Newfoundland and blocking patterns
over the North Atlantic (Shabbar et al., 2001), resulting in usually colder conditions in the eastern United States and warmer conditions in Greenland. Through a higher occurrence of low-pressure systems entering the Mediterranean basin, wetter conditions are more frequent than normal in southern Europe during a negative phase. While the link between the NAO and winter precipitations in northern Europe and in the Mediterranean region is moderately strong (Hurrell, 1995; Trigo et al., 2002), the effects of the NAO phase on precipitation and snowfall amounts in eastern North America are less clear.

Various studies have shown that a slight negative correlation exists between the winter NAO phase and total winter precipitation in New England (Bradbury et al., 2002a; Ning and Bradley, 2015). A significant negative correlation between NAO and seasonal snowfall in New England (Hartley and Keables, 1998; Bradbury et al., 2003) and in other parts of northeastern US has also been reported (Kocin and Uccellini, 2004; Notaro et al., 2006; Morin et al., 2008). Moreover, Huntington et al. (2004) found a significant negative correlation between the NAO and the snowfall-to-precipitation ratio
(S/P) in the northeastern US. Moving northward and westward from New England the correlation with snowfall gradually fade toward zero as shown by Ning and Bradley (2015). Kocin and Uccellini (2004) have studied the snowfall climatology of the Northeast urban corridor in depth and have noticed that winters with a low NAO almost always coincide with the high snowfall years. While the low-frequency variations in the NAO proved to be quite well correlated with seasonal snowfall in that region, their study also demonstrated that the occurrence of high-impacts winter storms is strongly linked to the daily
NAO value. A negative daily NAO was actually observed during the onset of most of the major snowstorms that affected the US east coast during the 1950-2000 period (Kocin and Uccellini, 2004). However, Archambault et al. (2008) and Notaro et al. (2006) have found that precipitation events frequency in the northeastern US is slightly higher in positive NAO conditions.

Several studies (Stone et al., 2000; Bonsal and Shabbar, 2008; Vincent et al., 2015; Whan and Zwiers, 2017) have
investigated the relationship between precipitation and teleconnection patterns in Canada and have found that winter precipitation is generally poorly correlated with the NAO in eastern Canada. An exception is northeastern Quebec and Labrador, where high NAO phases are associated with less precipitation than low NAO phases. Furthermore, Fortin and Hetu (2014) have found that the snowpack depth in the Chic-Chocs mountains (eastern Quebec) is very poorly correlated with the NAO. As winter snowfall largely contributes on the seasonal maximal snowpack depth, this result suggests that the
NAO may not have a significant impact on snowfall in that region.

The variability in winter precipitation and snowfall depending on the NAO can be explained by the shift in the storm track over North America and the North Atlantic. Bradbury et al. (2003) explained the negative correlation with



snowfall in New England with a southward shift of the prevailing storm track and colder temperatures during negative NAO winters. As almost half of the total precipitation during winter in the northeastern US is produced by coastal storms, known as Nor'easters, (Frankowski and DeGaetano, 2011), it is clear that the variability of winter precipitation in that region partly comes from the variability of the cyclogenesis and occurrence of these storms. Several studies have focused specifically on the storm track variability in the North Atlantic and showed that the negative phase of the NAO favors a more southern and zonal storm track offshore of the eastern US (Rogers, 1990; Serreze et al., 1997; Riviere and Orlanski, 2007), as well as an eastward displacement of the mean trough axis over North America (Bradbury et al., 2002b). The frequent occurrence of a blocking pattern over Greenland during NAO negative phases (Shabbar et al., 2001; Scherrer et al., 2006; Woollings et al., 2008) is often considered to be associated with this variability, as it strongly affects the circulation over the North Atlantic and adjacent regions. In a broader region, Wang et al. (2006) have found that a strong positive NAO is always associated simultaneously with more frequent cyclone activity in the high Canadian Arctic and less frequent activity on the east coast.

Although previous studies have delved into the link between the NAO and winter precipitation in eastern North America, few studies have provided clear and explicit explanations for these relationships. Therefore, in this study we looked at the effect of the NAO on various winter precipitations parameters (rainfall, snowfall, S/P and snow depth) over a wide domain covering southeastern Canada as well as the northeastern US. Furthermore, we use the recent ERA5 reanalysis dataset (1979-2018) to track extratropical cyclones, which account for almost all of the winter precipitation in eastern North America (Pfahl and Wernli, 2012), and to calculate the blocking frequency over the North Atlantic Basin. In doing so, we aim at providing an explanation for the precipitation variability based on the storm track variability associated with the phases of the NAO.

The paper is organized as follows. Data and methodology are described and discussed in section 2. Section 3 presents the results obtained from statistical analysis on the winter precipitations and track density variability with the NAO. The main findings are summarized in section 4, along with the conclusion.

## 2 Data and methods

### 2.1 Datasets

The NAOI used in this analysis is the monthly station-based index (Hurrell, 1995) based on the difference of normalized sea level pressure (SLP) between Lisbon, Portugal and Stykkisholmur/Reykjavik in Iceland (taken from the National Center for Atmospheric (NCAR) website: https://climatedataguide.ucar.edu/climate-data/hurrell-north-atlantic-oscillation-nao-index-station-based; Hurrell and National Center for Atmospheric Research Staff, 2020). Positive value of the NAOI is given by higher than average SLP over Portugal and lower than average SLP over Iceland. As in previous studies (Ning and Bradley, 2015; Bradbury et al., 2002a, Huntington et al., 2004), we consider the December to March period (DJFM), because consistent snowfall and winter conditions usually occurs during these 4 months over most of the study area. The monthly NAOI has been used to calculate the link between winter precipitations and the NAO.



The daily snowfall, total precipitation and snowpack depth in snow water equivalent (SWE) data comes from the European Center for Medium-range Weather Forecasts (ECMWF) ERA5 high resolution (0.25°) reanalysis dataset for the period 1979-2018 (Hersbach et al., 2020). The precipitation and snowfall totals used for calculating correlations are the summed total precipitation and snowfall for each month at every grid-points of the domain of interest. The snowfall-to-precipitation ratio (S/P) was calculated by dividing snowfall by total precipitation, as in Huntington et al. (2004). To analyze

how the annual maximum of snowpack depth varies depending on the winter (DJFM) NAO phase, the seasonal maximum of SWE were taken over the course of the snow year (August 1st to July 31th), as the maximum of snow depth is sometime achieved later than March in some parts of the domain. The winter climatology of total precipitation, snowfall, S/P and maximum seasonal SWE using the ERA5 dataset is shown in Fig. 1.

      Linear correlation coefficients with the monthly NAOI were first calculated for total precipitation, snowfall and S/P. For

the snowpack SWE, correlation coefficients were calculated between maximum SWE and seasonal (DJFM) NAO. Correlations were calculated using the Pearson R-value method at every grid-point. The statistical significance of these correlations using the p-value were determined at the 0.05 level. For statistical comparison between high NAO and low NAO months, the precipitation, snowfall and S/P were averaged for the 40 months with the highest NAO (1st quartile) and for the 40 months with the lowest NAO (4th quartile). For the snowpack maximum SWE analysis, the 10 winters with the highest

NAOI were compared to the 10 years with the lowest NAOI. To measure the statistical significance of these comparisons, the Student t test was used with a significance level of 0.05.

      Daily precipitation and snowfall data from the Global Historical Climatology Network (GHCN-daily) served as validation data for the same period as the ERA5 dataset (1979-2018). The data from 29 stations well distributed in the domain has been analyzed. Of these stations, 14 are located in eastern Canada, 6 are located in New England, and 9 are

located in the Mid-Atlantic region of the United States (Fig. 2). These stations were chosen based on the availability of data in the 1979-2018 period. Each station has near-complete precipitation data in this time period, which is essential for the comparison between results. Many stations that had data covering the whole time period of interest were rejected because of the high percentage of missing data during some years. Station inclusion criteria were intended to prevent stations with short, non-representative data records to bias the analysis. The same calculations on monthly precipitation, snowfall and S/P that have been done on ERA5 gridded data have been performed using the station-based data. Comparison between station-based

and reanalysis data also helped validating the ERA5 data on a regional scale.

## 2.2 Storm tracking algorithm

      In order to track individual cyclone trajectories on the ERA5 dataset, we developed a storm-tracking algorithm, which is based on previous methods, particularly the ones presented in Murray and Simmonds (1991, 2008) and Hanley and

Caballero (2012). The main features of the algorithm are described here below.

      We first use a Cressman filter (averaging grid-points within a 400 km radius circle) to filter out small-scale stationary local minima in the SLP field in the inter-mountain regions and retain only synoptic scale low-pressure systems.





Local minima in the SLP field (< 1013 hPa) are then taken as low-pressure centers for the tracking process. Cyclone trajectories construction is based on the previous/current/next position assumption (e.g. Ahrens; 2018): to forecast the second
position in a cyclone trajectory, the 500-hPa wind field is used; afterward, the location of future low-pressure centers are extrapolated by using the difference between the previous and current centers locations. Matches between low-pressure centers at each time step are attempted by looking at the nearest centers found from the predicted trajectories. If several matches are possible for a single trajectory, absolute departure from position, 850-hPa vorticity, and pressure with predicted value is used for matching low-pressure centers. After each complete storm trajectory, elimination criteria are applied in
order to remove the stationary orographic features and short-lived features from the list of tracks found. These criteria include a minimum lifetime of 48 hours and a minimum total displacement of more than 10°.

The track density, also known as cyclone occurrence or frequency, is defined in this study as the number of cyclones centers that passed within a radius of 200 km of a grid-point. Cyclogenesis density is defined here as the number of cyclone genesis (first points in trajectories) located within a radius of 200 km of a grid-point. The spatial distribution of both
track density and cyclogenesis density climatology (Fig. 3) are well in agreement with previous work that used different tracking methods (e.g. Neu et al., 2013). In particular, the main region of lee cyclogenesis east of the Canadian and Colorado Rockies are very well represented, as well as the areas of maximum coastal cyclogenesis due to the land-sea temperature contrast just offshore of Cape Hatteras in North Carolina (Fig. 3a). The values shown in Fig. 3 are used as reference for the statistical analysis discussed in section 3.5.

**2.3 Blocking frequency**

Blockings near Greenland (i.e. Greenland block) are responsible for deeper throughs over eastern North America (e.g. Resio and Hayden, 1975), which consequently have an influence on cyclogenesis and storm track. For that reason, we investigated the changes in blocking frequency during positive and negative NAO months over the North Atlantic Basin (30°N-70°N, 85°W-0°E). The analysis of the blocking frequency using the ERA5 dataset was performed using a bi-
dimensional index that identifies reversals in the meridional gradient of the 500-hPa geopotential height (Pausata et al., 2015; Anstey et al., 2013; Tibaldi and Molteni, 1990). At each grid-point of latitude $\phi$ and longitude $\lambda$, the northward and southward meridional gradients of geopotential heights are respectively estimated following Eq. 1 and Eq. 2:

$$\Delta_{N}(\phi, \lambda) = \frac{Z_{500}(\phi, \lambda) - Z_{500}(\phi - 15°, \lambda)}{15°}, \tag{1}$$

$$\Delta_{S}(\phi, \lambda) = \frac{Z_{500}(\phi + 15°, \lambda) - Z_{500}(\phi, \lambda)}{15°}, \tag{2}$$

A blocking event at a grid-point is diagnosed when two conditions are verified: (1) $\Delta_{N}(\phi, \lambda) > 0$, indicating a reversal of the climatological conditions with easterlies equatorward of the grid-point, and (2) $\Delta_{S}(\phi, \lambda) < -10 \, m/°$, indicating westerlies poleward of the grid-point. The blocking frequency (%) is then calculated by the number of timestep that a grid-point is "blocked" divided by the total number of timestep within a time period.



## 3 Results

In this section we first discuss the relationship between the NAO and total precipitation (Sect. 3.1), snowfall (Sect. 3.2) and S/P (Sect. 3.3): for each variable, the spatial pattern of correlations with the NAO index is discussed as well as the difference in monthly averages between high NAO and low NAO months. In Section 3.4, we focus on the link between seasonal NAO and maximum snowpack depth. Finally, in Section 3.5 we present the effect of the NAO on storm tracks and cyclogenesis in North America. As changes in storm tracking is a key driver of precipitation and snowfall variability, the

results are used to explain the findings presented in previous subsections.

### 3.1 Changes in total precipitation

       The coastal regions of New England and the Mid-Atlantic show slightly negative correlations between -0.10 and -0.20 (Fig. 4b), with coefficient values near -0.15. In relative terms, these coastal areas received up to 15-20% less precipitation during positive NAO compared to negative NAO months (Fig. 4a). When moving northward or westward from

the northeastern US coast, the results show a gradual reversal of correlation coefficient, which becomes positive in southern Newfoundland and in the Great Lakes region. Some areas of significant positive correlation are found in Ohio and southwestern Quebec.  In Atlantic Canada, the correlation coefficient is as high as 0.3 on the southern coast of Newfoundland. The correlation is instead strongly negative over the entire Labrador. In particular, the correlation coefficient lowers to under -0.5 in coastal Labrador, and positive NAO tend to bring 50% less total precipitation than negative NAO

months (Fig. 4b).

       All over the domain of interest, the station data matches well the ERA5 results. Only one station (Goose Bay) shows a difference larger than 0.1 between the correlation coefficient measured with reanalysis and observation. In the northeastern US, while showing no statistically significant results, the spatial pattern of correlation measured is consistent with results from previous studies (Bradbury et al., 2002a; Ning et al., 2015). Similarly, the results in Eastern Canada are

fairly consistent with those of earlier studies (e.g. Stone et al., 2000). Overall, the results suggest that the NAO has only a slight impact on total precipitation over most of the domain of interest, except for Newfoundland and Labrador.

### 3.2 Changes in snowfall

       When looking at snowfall (Fig. 4c,d), the relationship between snowfall and the NAO is much stronger than for total precipitation, especially for the northeastern US (Fig. 4c,d). In a wide corridor from southern Nova Scotia to the

southeast US, a significant negative correlation is shown together with notable decrease in snowfall during NAO positive. The decrease of snowfall during a positive phase of the NAO compared to a negative phase is as high as 50% in the Mid-Atlantic region. The negative correlation between snowfall and the NAO is statistically significant all over the northeastern US except at the northernmost parts of the region, which implies that high snowfall is associated with the negative phase of the NAO. This is consistent with previous studies that found out that major snowstorms that affected the Northeast US are



often associated with a negative NAO (Kocin and Uccellini, 2004; Hartley and Keables, 1998). Moving northward into southeastern Canada, the ERA5 results show correlations fading toward zero and becoming weakly positive in most of the region between 45°N and 50°N. In Newfoundland and Labrador, the snowfall relationship pattern with the NAO is similar to the one between precipitation and NAO. The positive correlation in southern Newfoundland and the negative correlation in Labrador are a little weaker than the one found for precipitation.

Station results generally agree well with ERA5 results, except in southern Quebec, where a slight negative correlation also exists with station results, although not with ERA5 results. This indicates a bias between ERA5 results and station results in this region. Overall, in eastern Canada only snowfall in Greenwood (Nova Scotia) and Sept-Iles (Quebec) shows significant correlations with the NAO (negative and positive, respectively). These results indicate that the NAO has a much larger impact on the variability of winter precipitations in the northeastern US compared to southeastern Canada.

### 3.3 Changes in snowfall to total precipitation ratio

The winter snowfall-to-precipitation relationship with the NAO is showing a very similar pattern in the northeastern US as the link between snowfall and NAO (Fig. 4c,d). As for snowfall, southern New England and the Mid-Atlantic region are area with the strongest signal between NAO and S/P. In the Mid-Atlantic region, there is a difference of more than -50% between positive and negative phases of the NAO. The results found in southern New England is consistent with Huntington et al. (2004), who also found a correlation of around -0.3 between winter NAO and S/P in this region. Everywhere within the northeastern US, positive NAO months tend to bring a lower S/P ratio, the only exception being northern Maine. Southern parts of eastern Canada also show a decrease in S/P during positive NAO months. In southern Nova Scotia, this decrease is as high as 0.05 (i.e. from 45% to 40% of S/P), and the negative correlation is significant. In Newfoundland and northeastern Quebec, the S/P ratio is higher during positive NAO. However, these northern regions typically witness almost only snowfall during the cold season (Fig. 1c), meaning that the absolute difference is quite low. The difference in the S/P results between northeastern US and Labrador show that the latitudinal gradient of average S/P is weaker during negative NAO conditions, indicating a larger latitudinal variability in the rain/snow boundary region.

The ERA5 and station-based results are consistent with each other. The only station that shows a notable difference between reanalysis and observation is the city of Buffalo in New York state.

### 3.4 Changes in annual maximum snow water equivalent

While the spatial distribution of the correlation coefficient between the NAO and SWE (Fig. 5) is less homogeneous, it is coherent with the results for the winter snowfall (Fig. 4c,d). This is not surprising as high snowfall is well correlated with maximum SWE in winter. Significant negative correlation is again shown in the Mid-Atlantic coastal region. As south of 40°N, snow that falls during storms usually stays on the ground only for few days, this suggests that there is an increase of heavy snowstorms affecting the Mid-Atlantic during negative NAO winters. In this region, snowstorms that dropped considerable amount of snow usually dictates the maximum SWE obtained on a particular year, rather than being





the result of an accumulation of several individual snow events over the course of the cold season. This supports Kocin and Uccellini's observations showing that historical large snowstorms in the Northeast US often occur during a negative NAO phase (Kocin and Uccellini, 2004). While it is not statistically significant, negative NAO winters tend to be linked to a 15-

30% increase in snow cover depth in New England compared to positive NAO. This explains the increase in avalanche activity witnessed in the White Mountains during negative NAO years, as noted by Martin et al. (2017). In eastern Quebec, there is a significant positive correlation between SWE and NAO, which is also consistent with the positive correlation with snowfall and S/P.

As the snowpack is strongly affected by temperature, negative correlations and increase of maximum SWE in the

northeastern US during negative NAO could also be partly explained by the colder conditions prevailing in that region during the negative phase (Fig. 6), which was also pointed out in previous studies (Notaro et al. 2006; Ning and Bradley, 2015). In similar manner, colder temperature witnessed during a positive NAO phase in eastern Quebec (e.g. Wettstein and Mearns, 2002) can explain the positive correlation between SWE and the NAO found in that region.

**3.5 Changes in cyclogenesis and storm tracking**

In order to better understand the causes of the precipitation and snowfall anomalies discussed in the previous sections, we analyze the changes in track density and cyclogenesis across North America associated with the two phases of the NAO.

The most notable result is the considerable increase in cyclone occurrence in the western North Atlantic just offshore of the northeastern US coast during a negative phase of the NAO (~0.5 more cyclone occurring per month) (Fig.

7b). While a decrease in cyclone occurrence is seen in that region during a positive phase (Fig. 7a), the anomaly is not as strong as during a negative phase. This implies that negative NAO conditions particularly favor an increase in frequency of Nor'easters, as previous studies suggested (Thompson and Wallace, 2001; Kocin and Uccellini, 2004; Roller et al., 2016). Near both Cape Hatteras and Cape Cod, which are two typical regions of coastal storms cyclogenesis on the US east coast (Davis et al., 1993), the cyclogenesis is strongly favored during negative NAO (Fig. 8b).

The mechanism responsible for such an increase of coastal cyclogenesis during a negative phase is likely the change in positioning of the upper level through. A through is more likely to be present over eastern North America, as seen in the strong positive correlation between the 500-hPa geopotential height and the NAO over the eastern US (Fig. 9; see also Bradbury et al, 2002b; Ning and Bradley, 2015). The exit region of the upper level through and jet streak is then more often overlapping the high temperature contrast region near the Gulf Stream (e.g. Davis et al., 1993; Hirsch et al. 2001), leading to

strong cyclogenesis. While their cyclogenesis is favored during a negative NAO phase, the spatial pattern of anomalies show that cyclones tend to take a more zonal and southern path in the Atlantic. Rogers (1990) and Serreze et al. (2002) also observed that during negative NAO phases, coastal storms often diverge from the continent near the 45th parallel and follow a track almost directly east, as opposed to continuing on a more typical northeastward trajectory following the coastline. This suggests that they tend to affect the region in the vicinity of their formation area, with strong onshore flow and heaviest



snowfall north of their path. But as they rapidly move offshore, their effects are not felt as much in areas further up the east coast. This explains why the Mid-Atlantic region and New England have slight negative correlations between NAO and precipitation, but not Atlantic Canada. The more zonal storm track of coastal storms during negative phases is caused by the much higher frequency of blockings near Iceland and Greenland (Fig. 10; see also Shabbar et al., 2001), forcing the storms to move south of it, as shown by the significant decrease in number of storms over Newfoundland and south of Greenland (Fig. 7b). In eastern Labrador Sea, just south of Greenland, there is as much as 1 less cyclone occurrence per month during negative phases of the NAO.

Another important aspect of the changes in the storm track is the southwest-northeast oriented dipole in the cyclone occurrence anomaly over the eastern half of the US during negative NAO months (Fig. 7b). This dipole is a result of a general southward shift in the storm track. As the position of the rain/snow boundary during winter storms is strongly linked to the location relative to the storm center (e.g. Donaldson and Stewart, 1989; Kocin and Uccellini, 2004), this shift explain the higher snowfall and S/P witnessed all over the northeastern US and Nova Scotia during negative NAO months.

Over Labrador, a lower (higher) cyclone occurrence during NAO positive(negative) relative to neutral (Fig. 7b) is consistent with the negative correlations found between precipitation and the NAO and between snowfall and the NAO. The higher occurrence of cyclones over Labrador and the western Labrador Sea during negative NAO may be also caused by the Greenland blocking high that often develops during negative phases (Fig. 10). While most of the coastal storms will take a zonal path well south of the blocking high (also in Rogers, 1990), some coastal storms, along with systems coming from the west, will be steered north, west of Greenland because of the anticyclonic circulation associated with the blocking high. Therefore, the results show a higher occurrence of cyclones in the western part of the Labrador Sea, bringing higher-than-normal precipitation to Labrador and southwestern Greenland, as also pointed out in Auger et al. (2017). As they take a more northern track, these low-pressure systems advect warm air in the region, lowering the S/P. Thus, the tripole pattern found in the western North Atlantic and Labrador Sea, as seen in the cyclone occurrence (Fig. 11) and precipitation (Fig. 4b) correlations plots, is likely the result of the disruption of the typical North Atlantic storm track by the more frequent Greenland blocking high during a negative NAO phase. The disruption forces cyclones to move one way or the other around the blocking high, depending of their position relative to the circulation patterns. On the contrary, positive NAO conditions lead to less storm track variability in the North Atlantic, and cyclones tend to follow in general more similar paths as they move over Newfoundland and head toward Iceland (Fig. 7a).

In western North America, a strong positive anomaly on lee cyclogenesis in Western Canada (Fig. 8a) gives way to an increase in cyclonic activity in Canada and the Arctic during a positive NAO phase. When these systems travel across the continent and bring precipitation in Eastern North America, they are called Clippers. Although they are rarely associated with extreme precipitation, there is an increase of their occurrence during positive NAO all over Canada, including Quebec and Atlantic Canada (Fig. 7a). This is coherent with Wang et al. (2006), who also found a positive correlation with cyclone occurrence in the Canadian Arctic. Contrary to the increase of lee cyclogenesis in Canada, there is a decrease in cyclogenesis in Colorado and Montana during positive NAO phases (Fig. 8a). The decrease in cyclogenesis in Colorado likely results in a



decrease in the occurrence of Colorado lows affecting eastern North America. The pattern of positive anomaly of
cyclogenesis in southern Alberta and negative in Montana during positive NAO is a sign of a northward shift of the Northern
Rockies storm track. This response could have a significant impact on the occurrence of storms in the Prairies region of
Canada, such as an increase of prairie blizzards frequency.

For eastern Canada, the increase of frequency of Alberta Clippers may be balanced by the decrease in frequency of
coastal storm, resulting in no significant anomaly of cyclone occurrence during a positive phase (Fig. 7a). However, coastal
storms and Colorado lows are known to be related to extreme precipitation as opposed to Alberta Clippers and Northern
Rockies lows that bring lighter precipitation in eastern Canada. This suggest that there is a decrease of days with extreme
precipitation during a positive NAO phase in Quebec and Atlantic Canada, as highlighted by Ning and Bradley (2015).

The spatial pattern of correlation between the NAO and track density (Fig. 11) are consistent with the anomalies of
cyclone occurrence for both positive/Neutral and negative/Neutral NAO. The large areas of significant negative correlation
in the Gulf/Atlantic storm track region and the significant positive correlation in Canada's Western and Arctic regions
further confirm the results of this analysis.

## 4 Summary and conclusions

In this study we examined the total precipitation, snowfall and snowfall-to-precipitation ratio (S/P) variability in
relation with the NAO during the winter season (December to March) over North America, with a particular focus on
northeastern US and southeastern Canada. In order to better understand these changes in precipitation and snowfall, we also
investigated the storm track and cyclogenesis variability associated with the NAO phases, as well as the changes in blocking
frequency across the North Atlantic Basin.

First, we show that while there is only a slight negative correlation between total precipitation and the NAO in the
northeastern US (Fig. 4b), the average snowfall is instead strongly affected by the NAO phase in that region (Fig. 4d). Both
reanalysis and station-based results show that positive NAO phases tend to bring considerably less snowfall compared to
negative NAO phases months over a wide region covering Nova Scotia, New England and the Mid-Atlantic of the United
States (Fig. 4c). Henceforth, a significant negative correlation is also seen between S/P and the NAO in the same region (Fig.
4f). These results are explained by the major increase (decrease) in cyclogenesis of coastal storms near the US east coast
during negative (positive) NAO phases (Fig. 8), as well as a southward (northward) shift in the mean storm track over the
United States (Fig. 7). The physical mechanism responsible for this increase in cyclogenesis is linked to the development of
a high-pressure system over Greenland that forms more (less) frequently during negative (positive) NAO phases (Fig 10;
Shabbar et al., 2001), resulting in a blocking in the North Atlantic and a deeper through over the eastern US (Fig. 9;
Thompson and Wallace, 2001). The position of the through and the cold air associated then favors the development of
storms on the east coast near the northern boundary of the gulf stream. The blocking in the North Atlantic also forces
cyclones to follow a more southerly and zonal storm track, leading to fewer cyclones tracking directly over New England



and Atlantic Canada (Fig. 7b). As a result, negative correlations between the NAO and precipitation, snowfall and S/P in coastal northeastern US slowly fades when moving northward 45° N. Moreover, positive NAO phases are linked to a considerable increase in lee cyclogenesis east of the Canadian Rockies compared to neutral NAO conditions (Fig. 7a). As cyclones that form due to lee cyclogenesis usually travel eastward across the continent to affect southeastern Canada, a slight

positive correlation is found between cyclone occurrence and the NAO over Quebec and Atlantic Canada. Even with the increase (decrease) of cyclone occurrence during NAO positive (negative) over southeastern Canada, the only regions where total precipitation is significantly positively correlated to the NAO phase are limited parts of eastern Quebec and southern Newfoundland (Fig. 4a). Over Labrador and the adjacent Labrador Sea, the correlation between cyclone occurrence and the NAO is negative (Fig. 10). The presence of the Greenland block during negative NAO phases forces some storms to move

northward west of Greenland, advecting warm air and precipitation to the region during negative NAO conditions. This explains the negative correlations between precipitation, snowfall and the NAO observed over Labrador (Fig. 4b,d).

To conclude, using a combination of station-based and high resolution ERA5 reanalysis data over a longer and more recent period (1979-2018), our study provides additional results on the mechanisms responsible for winter precipitations and storm track variability in the northeastern US and southeastern Canada compared to previous studies. In

particular, very few studies have investigated in detail the NAO impacts in southeastern Canada as done here. Given that global forecast models can predict reasonably well large-scale regimes such as the NAO at lead time of several weeks (e.g. Johansson, 2007; Scaife et al., 2014; Black et al., 2017), the results presented in this study could prove valuable for improving the sub-seasonal forecasting of the probability of the occurrence of precipitation events, in particular heavy snowfall events.


**Data availability**. ERA5 reanalysis data are provided by the European Centre for Medium-Range Forecasts (https://www.ecmwf.int/en/forecasts/datasets/reanalysis-datasets/era5; Hersbach et al., 2020). The NAO index data are provided by the National Center for Atmospheric Research (https://climatedataguide.ucar.edu/climate-data/hurrell-north-atlantic-oscillation-nao-index-station-based; Hurrell and National Center for Atmospheric Research Staff, 2020).


**Author contributions**. JC performed most of the analysis and the writing of the manuscript. FSRP supervised the project, and helped with the analysis design, results interpretation, and manuscript writing.

**Competing interests**. The authors declare that they have no competing interests.


**Acknowledgements**. FSRP acknowledges financial support from the Natural Sciences and Engineering Research Council of Canada (grant RGPIN-2018-04981) and the Fonds de Recherche du Quebec—Nature et Technologies (2020-NC-268559). We acknowledge the European Centre for Medium Range Weather Forecasts (ECMWF) for providing access to ERA-5





reanalysis data. Additionally, we thank Katja Winger for helping in the design of the storm tracking algorithm and Gabriele
Messori for helpful comments on the manuscript.

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




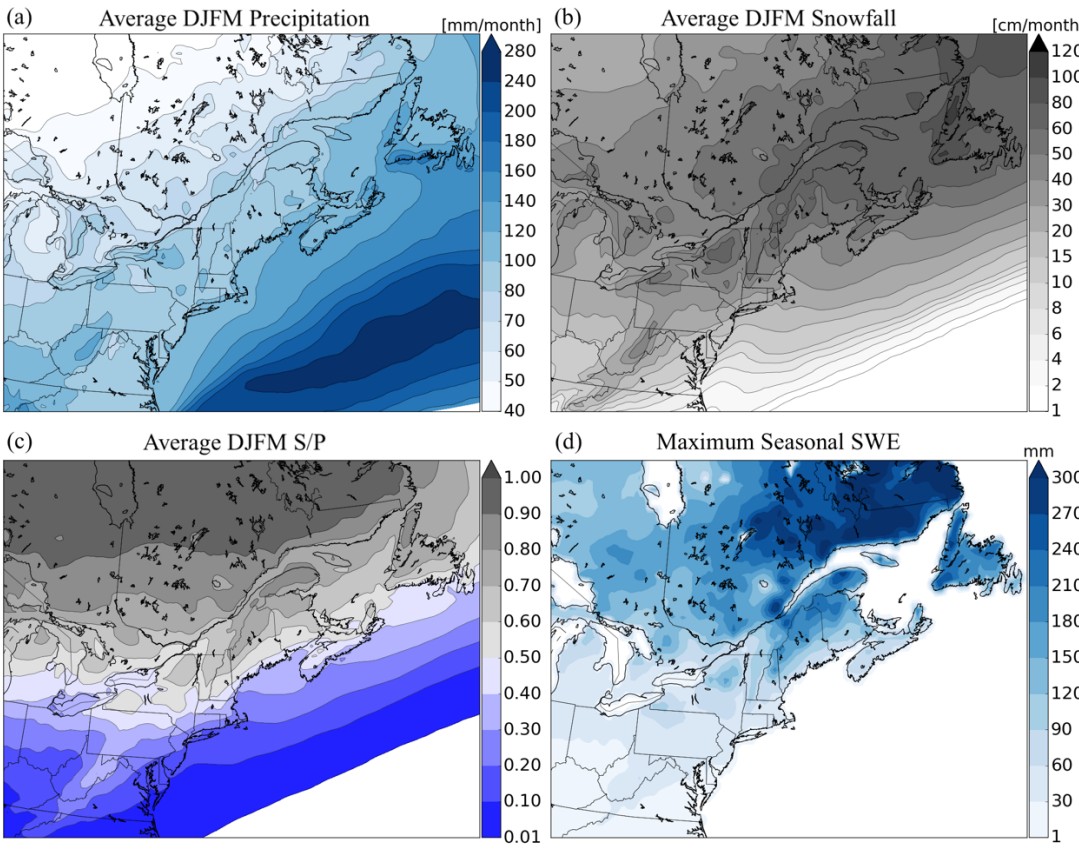

**Figure 1.** ERA5 1979-2018 climatological values of winter (December to March, DJFM) (a) total precipitation, (b) snowfall and (c) snowfall-to-precipitation ratio. (d) Average maximum SWE of the snowpack over the course of a snow year (August 1st-July 31th).

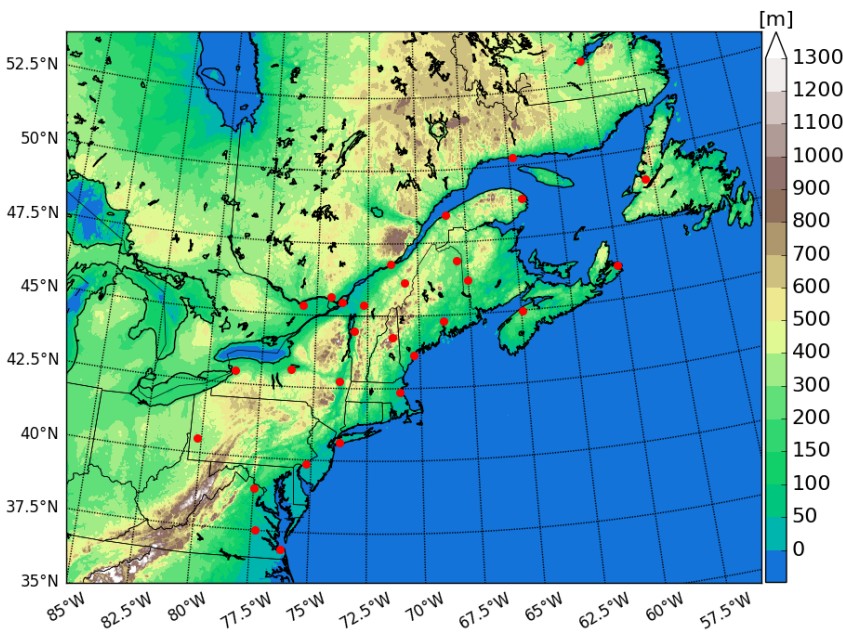


**Figure 2.** Map of the domain of interest with elevation included. Red dots represent the locations of the weather stations used in the study as data validation. The elevation data shown is ETOPO1 1 Arc-Minute Global Relief (retrieved from https://www.ngdc.noaa.gov/mgg/global/; Amante and Eakins, 2009).

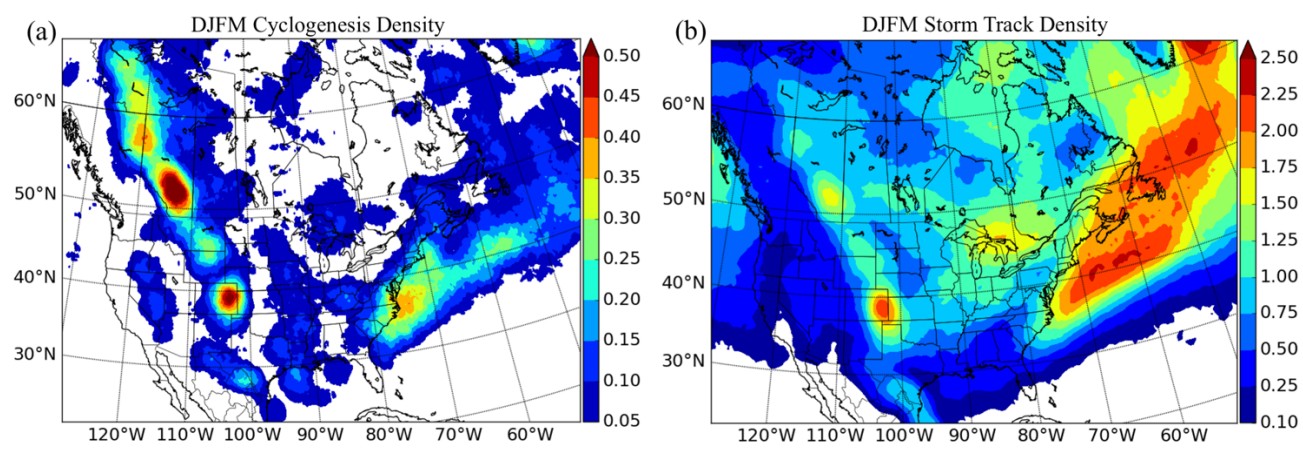


**Figure 3.** ERA5 average cool-season (December to March) (a) cyclogenesis density and (b) storm track density (or cyclone occurrence) over the 1979-2018 period. The units are cyclones per month within 200 km radius.



**Figure 4.** (a) Relative difference between average precipitation during positive NAO and negative NAO months (40 months composites, 1st and 4th quartiles) in winter (DJFM), positive values mean more precipitation during high NAO. (b) Correlation coefficients between monthly precipitation and monthly NAOI. Purple contours represent the 0.05 statistical significance level threshold. Shaded circles represent station-based correlation coefficient that are statistically significant at the 0.05 level, and diamonds represents values that are not statistically significant. (c), (d) as in (a) and (b), but for the relationship between monthly NAO and snowfall. (e), (f) as in (a) and (b), but for the relationship between monthly NAO and S/P.




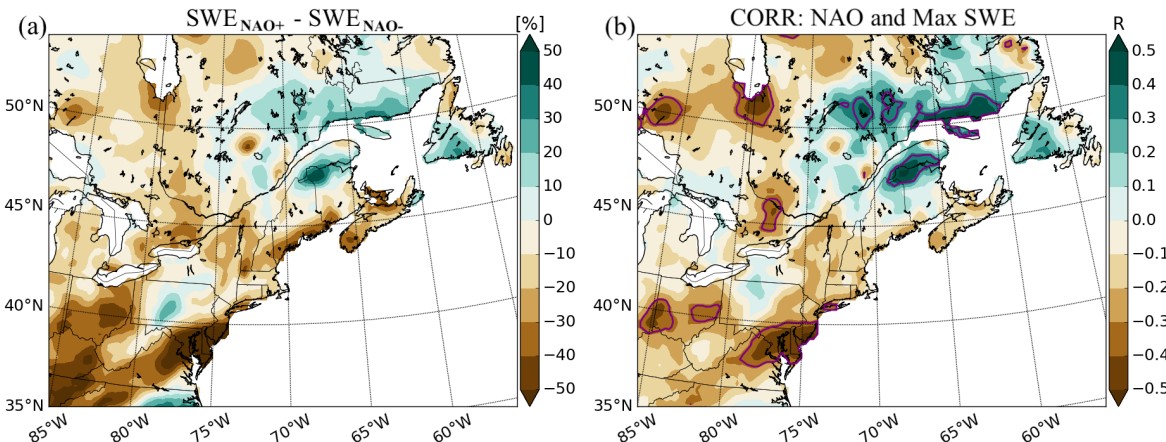

**Figure 5.** (a) Relative difference between average maximum snow depth in Snow Water Equivalent (SWE) during positive NAO and negative NAO winters (10 winters composites, 1st and 4th quartile). Positive values mean more SWE during high NAO. (b) Correlation coefficients between DJFM NAOI and maximum seasonal SWE. Purple contours represent the 0.05 statistical significance level threshold.


**Figure 6.** (a) Relative difference between average temperature during positive NAO and negative NAO months (40 months composites, 1st and 4th quartiles) in winter (DJFM), positive values means warmer temperatures during high NAO. (b) Correlation coefficients between monthly temperature and monthly NAOI. Purple contours represent the 0.05 statistical significance level threshold.

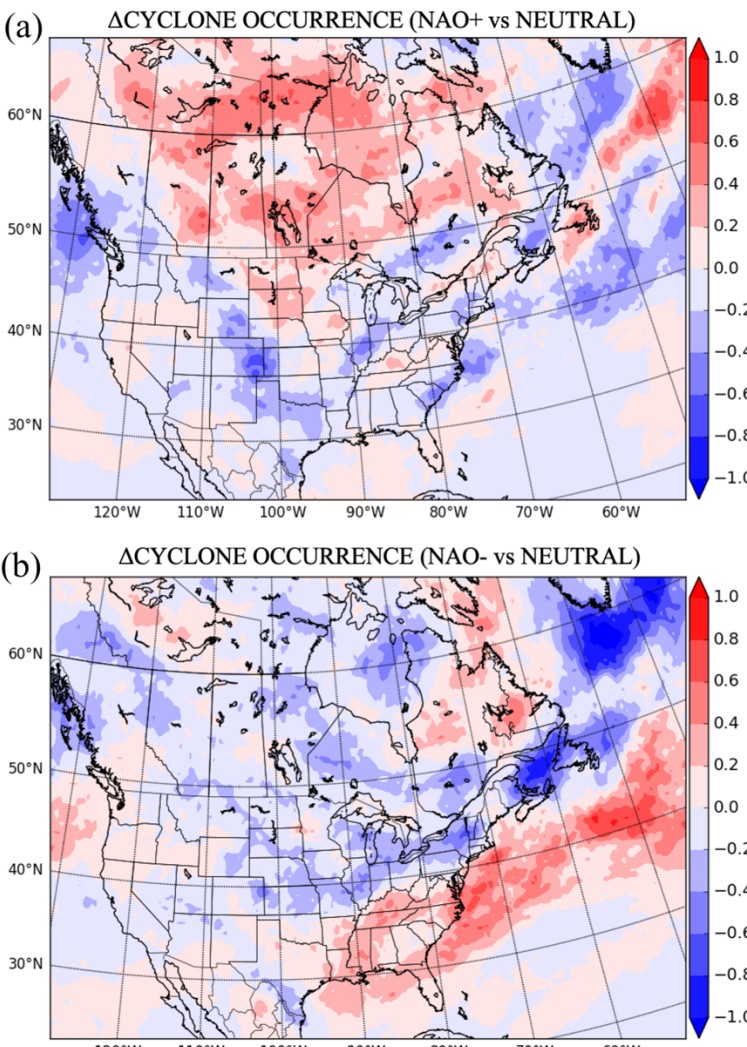


**Figure 7.** Anomalies of monthly track density (cyclone/month within 200 km radius) over the course of (a) positive NAO and negative NAO months (40 months composites, 1st and 4th quartiles) in winter (DJFM), positive values mean higher cyclone occurrence. Anomalies are calculated in relation to an average neutral NAO month.

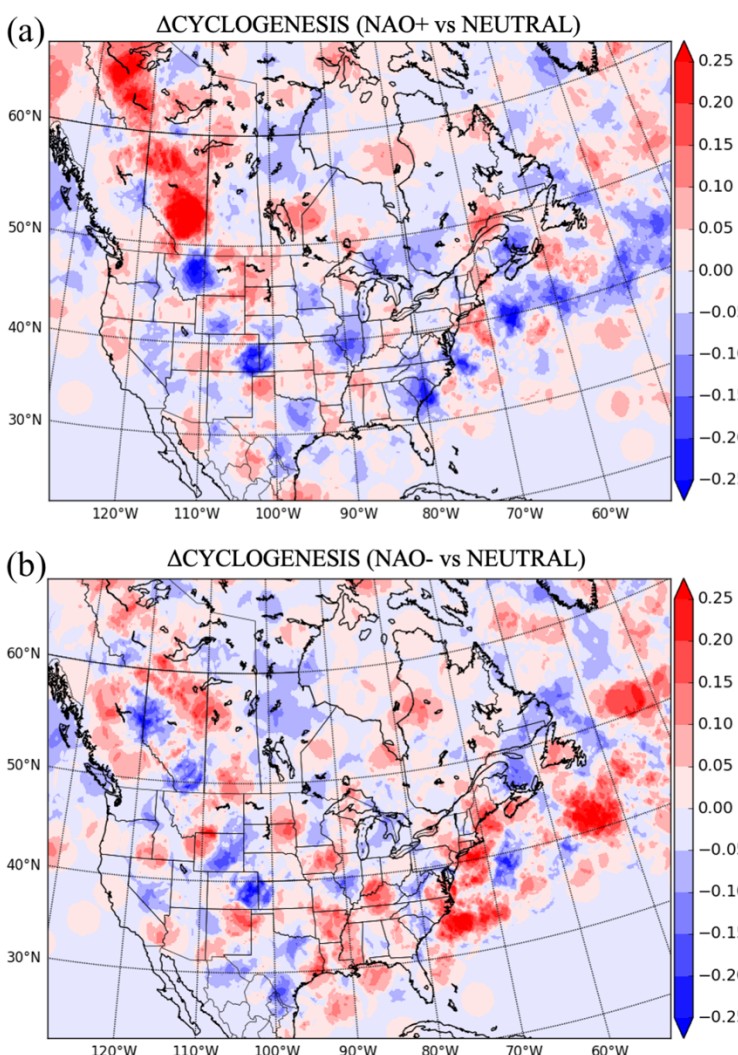

**Figure 8.** Anomalies of cyclogenesis density (cyclogenesis/month within 200 km radius) over the course of (a) positive NAO and negative NAO months (40 months composites, 1st and 4th quartiles) in winter (DJFM), positive values mean higher cyclogenesis. Anomalies are calculated in relation to an average neutral NAO month.

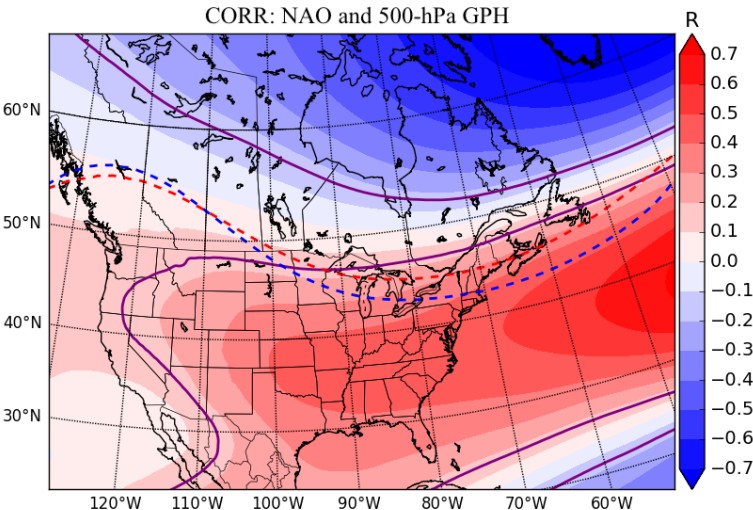

**Figure 9.** Correlation coefficients between monthly 500-hPa geopotential heights and monthly NAOI in winter (DJFM). Purple contours
represent the 0.05 statistical significance level threshold. Dashed lines represent the average position of the 5400 m geopotential height
isohypse during average low NAO months (blue) and average high NAO months (red).

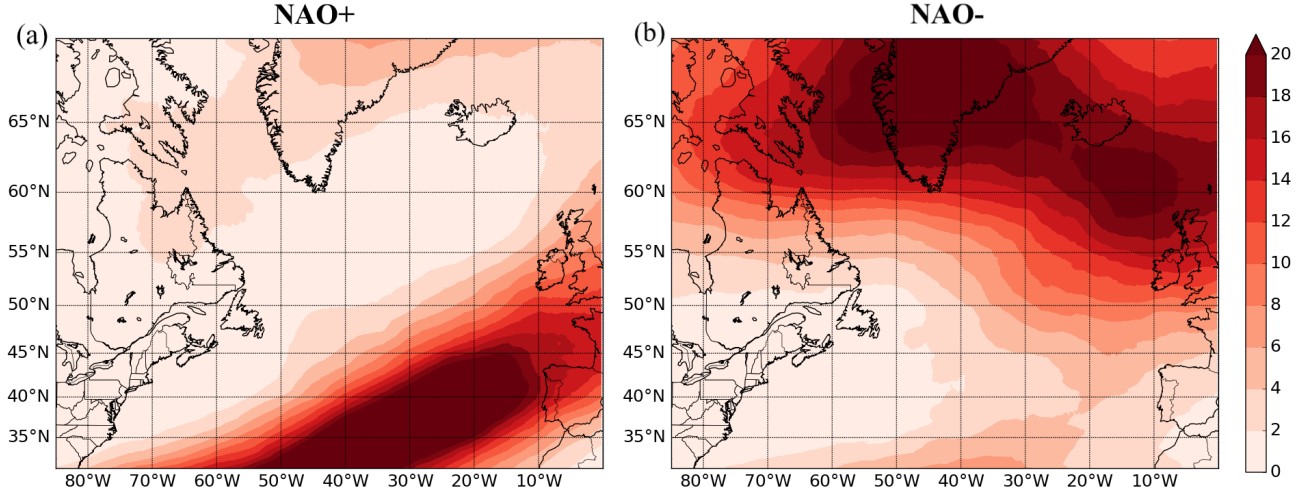

**Figure 10.** Blocking frequency averaged over (a) positive NAO months and (b) negative NAO months (40 months composites, 1st and 4th
quartiles) in winter (DJFM).

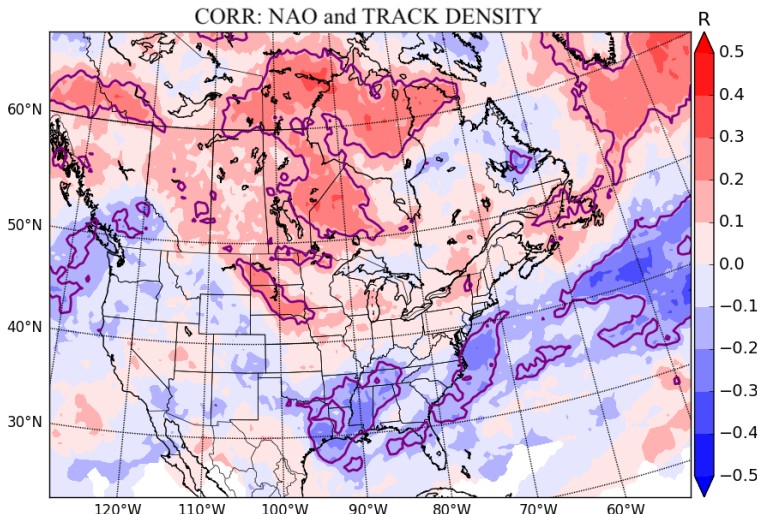

**Figure 11.** Track density correlation with the monthly NAOI in winter (DJFM). Purple contours represent the 0.05 statistical significance
level threshold.