# Peer review of "Impacts of the North Atlantic Oscillation on Winter Precipitations and Storm Track Variability in Southeast Canada and Northeast US"

_Weather and Climate Dynamics, 2020_

## Short Comment (SC1) · 16 Jun 2020

Short reply to Reviewer #1

We would like to thank Reviewer #1 for the insightful comments, which we believe, can substantially improve our manuscript. We provide this preliminary reply to clarify some of the aspects that did not fully convince the Reviewer. We feel that we are able to address all of the Reviewer's major concern, and we think this reply may provide useful information for the other reviewer(s) as well, who may have similar concerns. This reply

is not meant to be the full reply to Reviewer #1, which will be submitted after receiving all reviewers' comments.

Reviewer #1's major concern was the lack of the novelty of our study. We acknowledge that the novelty of the paper was not properly conveyed. In the revised manuscript, we will make sure to highlight the difference and the added value of our results relative to the existing literature, as well as to make the purpose of the study much clearer.

Our manuscript stands out by offering a comprehensive overview of the influence of the NAO on the winter climate of eastern North America specifically. A particularity of our study is that we used individual tracks of low-pressure systems to analyse the storm track variability throughout the North American continent and the western North Atlantic Ocean. Therefore, we clearly and explicitly show the regional anomalies in cyclogenesis and storm tracks associated with the phases of the NAO, which are then responsible for changes in precipitation and snowfall. To our knowledge, this was not performed in previous studies.

Indeed, several studies in the cited literature discussed the relationship between mean precipitation, snowfall and the NAO in similar domains of interest. The novelty in our study for these analyses is the use of recent reanalysis data instead of station data. We will however present these more briefly in the revised version, as we will focus our discussion on the more novel results. Furthermore, we will follow the reviewer #1 suggestion and in the revised manuscript we will include the analysis of extreme precipitation and snowfall to provide further novelty. We feel instead that the analysis of weather regimes would be redundant with a recently published study (Roller et al, 2016).

Finally, we will also include a more in-depth validation of our tracking algorithm including the overall climatology over a larger domain and how it tracks specific cases. As suggested, we will make direct comparisons with the tracking algorithms presented in Neu et al. (2013) using the same units.

[Figure]

Of course, all of the Reviewer's other comments and suggestions will be taken into account while making changes to the manuscript.

Again, we appreciate the comments of the Reviewer #1, which provided this opportunity to clarify the novelty of our study and outline the steps we will do to make improvements to our paper.

References

Neu, U., Akperov, M. G., Bellenbaum, N., Benestad, R., Blender, R., Caballero, R., et al.: IMILAST: A community effort to intercompare extratropical cyclone detection and tracking algorithms, Bulletin of the American Meteorological Society, 94(4), 529-547, doi:10.1175/bams-d-11-00154.1, 2013.

Roller, C. D., Qian, J. H., Agel, L., Barlow, M., and Moron, V.: Winter weather regimes in the northeast United States, Journal of Climate, 29(8), 2963-2980, doi:10.1175/jcli-d-15-0274.1, 2016.

---

## Referee Comment (RC2) · Anonymous Referee #2 · 9 Jul 2020

Chartrand and Pausata present a study into teleconnections between the North Atlantic Oscillation and wintertime precipitation and storm track variability across southeastern Canada and northeastern U.S. The authors utilize ERA5 and station observations to perform multiple tests including correlation and storm track analysis. They conclude 1) that positive (negative) NAO anomalies are associated with 1) reduced (increased) snowfall from the Mid-Atlantic U.S. to Nova Scotia CA, and 2) decreased (increased) coastal storm cyclogenesis in the vicinity of the U.S. East Coast, which is evidenced from a negative correlation between snow/precipitation ratio and the NAO index over

the aforementioned region.

While I think this study has potential, the manuscript needs to be better focused and streamlined before publishing. In particular, in its current form the manuscript reads more like a review than a research paper, and I am still left wondering what is new and what has been previously reported. Along these lines, I recommend shortening the introduction, keeping only what is absolutely necessary background (e.g., the first sentence cites five previous studies in supportive of the definition of the NAO – are all of these critical?), and making a more explicit statement of motivation. In the results section, report only the results, and move interpretations and comparisons against previous work to a discussion section.

Otherwise, this additional insight to into the NAO association with winter storm tracks and snowfall distribution across the study region is valuable and should be of interest to readers.

---

## Author Comment (AC1) · 10 Oct 2020

Impacts of the North Atlantic Oscillation on Winter Precipitations and Storm Track Variability in Southeast Canada and Northeast US

**Response to Reviewers**

We would like to thank both reviewers for their time and effort to review our manuscript. We have addressed to the best of our knowledge all major and minor comments raised by the reviewers. In doing so, we feel we have crafted a revised manuscript that is more rigorous in content, and better presents the results of the study. Here we list the main changes that we have made:

- We now include an analysis of extreme precipitation and snowfall, rather than only mean precipitation and snowfall amounts. We have added a new subsection (Sect. 3.2) and a new figure, which present these results.
- We have written a new discussion section (Sect. 4), where we have moved all interpretation, discussion and comparisons to previous studies. We have therefore carefully re-written the results section (Sect. 3) to present only the most important and novel results.
- Throughout the manuscript, we have made the purpose of the manuscript clearer.
- The introduction and conclusion were made shorter.
- We have made the explanation of the storm tracking algorithm more detailed.

Below, we copy the reviewers' comments in bold and describe how each of these issues has been addressed in the revised manuscript. The revised version of the manuscript is attached after the answers to reviewers' comments, and the changes compared to the original version are highlighted in bold.

**Response to reviewer #1**

In the manuscript "Impacts of the North Atlantic Oscillation on Winter Precipitations and Storm Track Variability in Southeast Canada and Northeast US" the authors present a composite analysis of snowfall, total precipitation and cyclone track densities. They focus their discussion on the northeasternmost part of the American continent. The manuscript is easy to follow and the Figures are generally clear. I do however have considerable doubts about the novelty of the findings in this article. The authors cite quite a few studies in the introduction and throughout the manuscript that considered similar diagnostics with a similar scientific question for a similar region. Not surprisingly, throughout the manuscript, the authors then describe their findings as consistent with what has been pointed out before. That is in my eyes not enough to warrant a new publication. I am nevertheless recommending major revisions in the faith that the authors will be able to derive genuinely novel results from a similar set of results to the one presented in the current manuscript. It might however require a shift or broadening of the scope of the manuscript. For example the authors could explicitly consider extreme precipitation and snowfall rather than cold season means, or relate precipitation and snowfall to weather regimes (such as Greenland blocking) rather than the NAO. Whatever the authors' choice, the motivation and purpose of the study must become much more clear, in particular in the introduction and conclusion.

We thank the reviewer for the thorough evaluation of the manuscript and appreciate the positive remarks on the structure of the manuscript and the general clarity of the figures.

We thoroughly went through the literature in order to better compare our results to earlier studies that had similar scientific questions, highlighting throughout the manuscript similarities in our results relative to other studies. We believe that this may have given the impression of a lack of novelty in our manuscript.

However, our manuscript brought many new results, which we believe will be of scientific interest. Particularly, we used reanalysis data to study the relationship between snowfall and the NAO and did a full investigation of the impact of the NAO on the winter climate of eastern Canada, which were not done before. While the precipitation and snowfall variability with the NAO in the eastern US has been studied before, we wanted to add physical explanations to these

relationships. For example, the study of the storm track variability with the NAO over North America is one of the novel aspects of our manuscript and following the reviewer's suggestion, we have also included an analysis of extreme snowfall and precipitation events. Finally, we have also rewritten the introduction and the conclusion to better highlight the motivation and the purpose of the study.

The change in extreme precipitation and snowfall has been analyzed by using various extreme climate indices; the number of days with precipitation above 10 mm (R10mm), the number of days with precipitation above 20 mm (R20mm), the maximum daily precipitation (Rx1day), and the maximum 5-days precipitation (Rx5days). However, as the results are fairly similar among these indices (see Fig. R1), we only show in the revised manuscript the results for the relationship between the NAO and R10mm.